# HER2-targeting antibody drug conjugate FS-1502 in HER2-expressing metastatic breast cancer: a phase 1a/1b trial

Qiao Li[1], Ying Cheng [2], Zhongsheng Tong[3], Yunjiang Liu[4], Xian Wang[5], Min Yan [6], Jianhua Chang[7], Shusen Wang [8], Caiwen Du[9], Liang Li[9], Chunjiao Wu[2], Mingxia Wang[10], Zhuo Wang[5], Zhuli Wu[11], Xingli Wang[11], Yongli Jin[12], Lei Diao[12], Yi Sun[12], Yongjiao Zhang[12], Ai-Min Hui[13] & Binghe Xu [1] ✉

Currently approved HER2-targeting antibody-drug conjugates (ADCs) for HER2-positive breast cancer (BC) are associated with safety concerns. In this multicenter, single-arm, dose-escalation (phase 1a) and dose-expansion (phase 1b) phase 1 trial (NCT03944499), patients with HER2-expressing advanced solid tumors received FS-1502 (an anti-HER2 ADC) with a 3 + 3 design in phase 1a; patients with metastatic HER2-positive BC received FS-1502 at the recommended phase 2 dose (RP2D) in phase 1b. The primary end points were dose-limiting toxicities (DLTs), maximum tolerated dose (MTD) and RP2D for phase 1a and objective response rate (ORR) for phase 1b. A total of 150 patients with HER2-expressing solid tumors (*n* = 5) and BC (*n* = 145) were enrolled (female, *n* = 146, 97.3%). One DLT each was reported at 3.0 and 3.5 mg/kg; the MTD was not reached. The RP2D was 2.3 mg/kg once every 3 weeks. Five (3.3%) patients experienced pneumonitis; four (2.7%) had grade 3 reversible ocular events. Of 67 HER2-positive BC patients receiving the RP2D, the best ORR was 53.7% (95% CI, 41.1-66.0%), including PRs confirmed (confirmed ORR, 37.5%) and pending for confirmation. FS-1502 was well tolerated with limited ocular and pulmonary findings and demonstrated promising antitumor activity in HER2-positive BC patients.

Human epidermal growth factor receptor 2 (HER2) overexpression has been observed in approximately 20% of breast cancer cases, and numerous other solid tumors with a frequency ranging from 2% to more than 50%, including gastric cancer, biliary tract cancer, colorectal cancer, non-small-cell lung cancer, uterine cancer and bladder cancer[1].

HER2 is a prognostic biomarker of breast cancer, with HER2 overexpression/amplification associated with poorer outcomes and higher rates of disease relapse and mortality versus HER2-negative cases[2–4].

Several HER2-targeted therapies have shown efficacy in clinical trials and are approved for the treatment of breast cancer and gastric

[1]Department of Medical Oncology, State Key Laboratory, National Cancer Center/Cancer Hospital Chinese Academy of Medical Sciences and Peking Union Medical College, Beijing, China. [2]Department of Oncology, Jilin Cancer Hospital, Changchun, China. [3]Department of Breast Medical Oncology, Tianjin Medical University Cancer Institute and Hospital, Tianjin, China. [4]Department of Breast Center, The Fourth Hospital of Hebei Medical University, Shijiazhuang, China. [5]Department of Medical Oncology, Sir Run Run Shaw Hospital, Zhejiang University School of Medicine, Hangzhou, Zhejiang, China. [6]Department of Breast Medicine, Henan Cancer Hospital, Zhengzhou, Henan, China. [7]Department of Medical Oncology, Cancer Hospital Chinese Academy of Medical Sciences, Shenzhen Center, Shenzhen, China. [8]Sun Yat-sen University Cancer Center, Guangzhou, Guangdong, China. [9]Department of Medical Oncology, Meizhou People's Hospital, Meizhou, Guangdong, China. [10]Department of Clinical Pharmacology, The Fourth Hospital of Hebei Medical University, Shijiazhuang, China. [11]Shanghai Fosun Pharmaceutical Industrial Development Co., Ltd., Shanghai, China. [12]Shanghai Fosun Pharmaceutical Development Co., Ltd., Shanghai, China. [13]EnCureGen Pharma, Guangzhou, China. ✉e-mail: xubinghe@medmail.com.cn

cancer, including monoclonal antibodies, tyrosine kinase inhibitors and antibody-drug conjugates (ADCs)[5-11]. Among the HER2-targeted therapies, ADCs are a rapidly growing area in cancer treatment[12]. An ADC is composed of a cytotoxic chemotherapeutic drug that is linked to a cancer-specific monoclonal antibody[13]. The anti-HER2 ADC binds to the HER2 antigen on the cancer cell surface, is internalized by cancer cells, and the cytotoxic payload is released to kill the target cancer cells after degradation by lysosomes[13]. Among 13 approved ADCs on the market worldwide, three are HER2-targeting ADCs, including fam-trastuzumab deruxtecan (T-DXd), ado-trastuzumab emtansine (T-DM1), and disitamab vedotin[14,15]. T-DXd is a HER2-directed antibody and topoisomerase inhibitor conjugate and has been approved for the treatment of adult patients with unresectable or metastatic HER2-positive breast cancer who have received a prior anti–HER2-based regimen either in the metastatic setting or in the (neo)adjuvant setting and have developed disease recurrence during or within 6 months of completing therapy; it has also been approved for the treatment of adult patients with unresectable or metastatic HER2-low (IHC 1+ or 2+/ISH-) breast cancer who have received prior chemotherapy in the metastatic setting or developed disease recurrence during or within 6 months of completing adjuvant chemotherapy[16]. T-DM1, a HER2-targeted antibody and microtubule inhibitor conjugate, has been approved for the treatment of patients with metastatic HER2-positive breast cancer who previously received trastuzumab and a taxane, separately or in combination, where patients should have either received prior therapy for metastatic disease, or developed disease recurrence during or within 6 months of completing adjuvant therapy; T-DM1 has also been approved for the adjuvant treatment of patients with HER2-positive early breast cancer who have residual invasive disease after neoadjuvant taxane- and trastuzumab-based treatment[17,18]. However, safety concerns with the approved anti-HER2 ADCs, such as interstitial lung disease (ILD)/pneumonitis, ocular and severe gastrointestinal toxicities still exist[19].

FS-1502 is a HER2-targeting ADC comprising a cancer-selective cleavable β-glucuronide linker, an anti-HER2 antibody derived from trastuzumab, and an antimitotic agent, monomethyl auristatin F (MMAF), which inhibits tubulin polymerization. The unique site-specific chemoenzymatic conjugation method used to attach the linker payload to the antibody enables the production of a homogeneous ADC with a defined drug-to-antibody ratio, increased linker stability compared to traditional approaches, and favorable physio-chemical properties[20]. The β-glucuronide cleavable linker technology results in tumor-selective release and activation of the payload while sparing toxicity to normal tissues[21,22]. FS-1502 exhibited more potent cell cycle arrest and target-dependent cytotoxicity than T-DM1 in vitro[22]. FS-1502 was found to target and accumulate in the HER2-positive breast cancer cell (JIMT-1) without accumulation in organs in a xenograft mouse model[23]. FS-1502 showed potent antitumor activity in xenograft models bearing HER2-positive breast cancer or gastric cancer tumors, and it could also inhibit the growth of HER2-low tumors that were resistant to T-DM1 in xenograft models[24].

This phase 1a/b study consisted of a dose-escalation part aimed to evaluate the safety and tolerability of FS-1502 in patients with HER2-expressing advanced solid tumors and a dose-expansion part aimed to evaluate the efficacy and safety of FS-1502 in patients with HER2-positive metastatic breast cancer.

## Results
### Patients and treatment

Patients were enrolled between November 11, 2019, and December 13, 2022. The cutoff date for safety and efficacy data was December 24, 2022, and for pharmacokinetic (PK) data was July 30, 2022. A total of 150 patients were enrolled and treated in this phase 1 trial (Fig. 1) at doses from 0.1 mg/kg every 4 weeks (Q4W) to 3.5 mg/kg every 3 weeks (Q3W) (both Q3W and Q4W in 1.0 mg/kg, 1.3 mg/kg and 1.7 mg/kg), with 89 patients treated at RP2D. There were 85 patients enrolled in phase 1a and 65 patients in phase 1b. The median study follow-up time was 5.8 months. Baseline characteristics of patients at RP2D and total are shown in Table 1 (baseline characteristics of all dose groups are listed in Supplementary Table 1).

For the whole study, the median age was 52.0 years (range 27-76), most patients were female (n = 146, 97.3%), and 145 (96.7%) patients were diagnosed with breast cancer; two (1.3%) had lung cancer, and one (0.7%) each had ampullary cancer, submandibular gland malignant tumor, and gastric cancer. For patients with breast cancer, 112 (77.2%) were HER2-positive and 32 (22.1%) had tumors with HER2-low expression. For non-breast cancer patients, two (40%) were HER2 IHC 2+ and three (60%) were HER2 IHC 3+. The most common metastatic sites

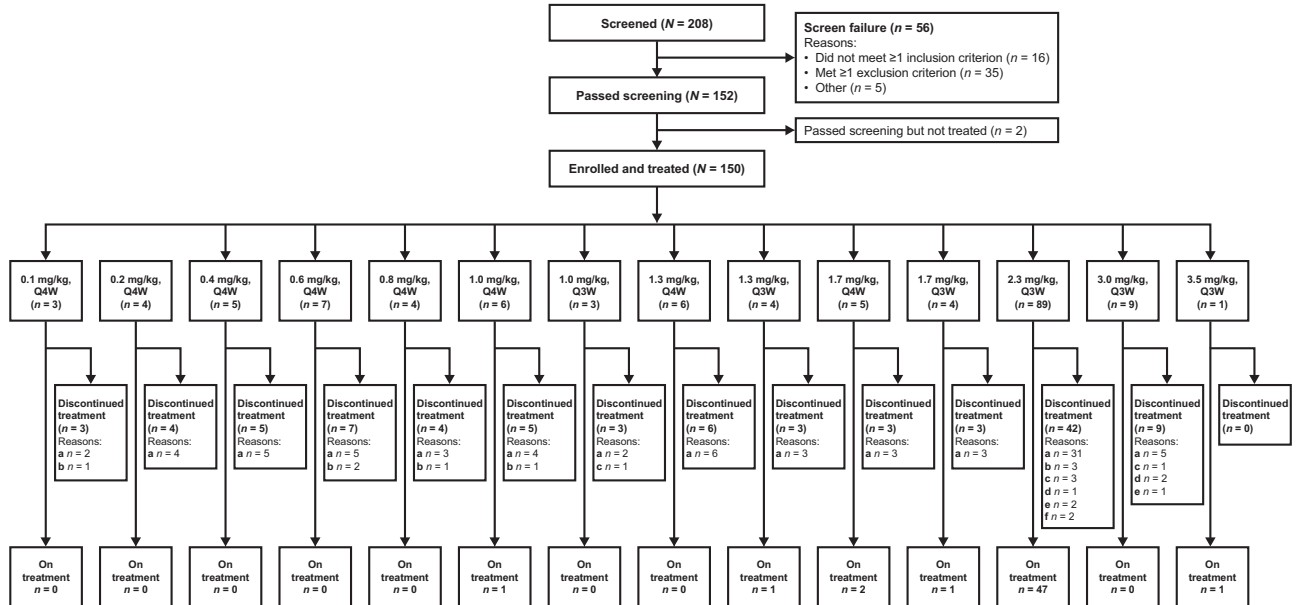

**Fig. 1 | Patient dispositions.** [a]Progressive disease. [b]Patient withdrew informed consent. [c]Adverse event. [d]Study termination by the investigator. [e]Death. [f]Other. *Q3W* once every 3 weeks, *Q4W* once every 4 weeks.

**Table 1 | Demographic and baseline characteristics**

| Dose | 2.3 mg/kg Q3W (n = 89) | Total (n = 150) |
|---|---|---|
| Median age (range), years | 52.0 (27-76) | 52.0 (27-76) |
| ECOG PS, n (%) | | |
| 0 | 41 (46.1) | 76 (50.7) |
| 1 | 48 (53.9) | 74 (49.3) |
| Tumor type, n (%) | | |
| Breast cancer | 87 (97.8) | 145 (96.7) |
| Others | 2 (2.2) | 5 (3.3) |
| Metastatic site, n (%) | | |
| Bone | 39 (43.8) | 63 (42.0) |
| Liver | 42 (47.2) | 63 (42.0) |
| Lung | 46 (51.7) | 81 (54.0) |
| Brain | 7 (7.9) | 11 (7.3) |
| Lymph nodes | 49 (55.1) | 91 (60.7) |
| Other | 26 (29.2) | 51 (34.0) |
| HER2 expression status, n (%) | | |
| HER2-low expression (HER2 1+ or HER2 2+/FISH-) | 16 (18.0) | 32 (21.3) |
| HER2-positive (HER2 3+ or HER2 2+/FISH+) | 71 (81.6) | 112 (74.7) |
| HER2 2+ | 1 (1.1) | 2 (1.3) |
| HER2 3+ | 1 (1.1) | 3 (2.0) |
| History of prior treatment lines, median (min–max) | 3.0 (1-16) | 3.0 (1-16) |
| History of anti-HER2 therapy for HER2-positive breast cancer pts, n (%) | 71 (79.8) | 112 (74.7) |
| Trastuzumab[a] | 69 (97.2) | 107 (95.5) |
| Pyrotinib[a] | 54 (76.1) | 75 (67.0) |
| Pertuzumab[a] | 28 (39.4) | 38 (33.9) |
| Lapatinib[a] | 14 (19.7) | 28 (25.0) |
| Inetetamab[a] | 12 (16.9) | 13 (11.6) |
| Other[a] | 6 (8.5) | 8 (7.1) |
| T-DM1[a] | 4 (5.6) | 6 (5.4) |

[a]The denominator is the number of patients with HER2-positive breast cancer in each dose group.

**Table 2 | Summary of adverse events (safety set)**

| Dose | 2.3 mg/kg Q3W (n = 89) | Total (n = 150) |
|---|---|---|
| Study drug-related TEAEs ≥15% for all patients by preferred term | | |
| AST increased | 66 (74.2) | 100 (66.7) |
| Hypokalemia | 59 (66.3) | 77 (51.3) |
| ALT increased | 39 (43.8) | 66 (44.0) |
| Proteinuria | 36 (40.4) | 51 (34.0) |
| Dry mouth | 32 (36.0) | 50 (33.3) |
| Platelet count decreased | 31 (34.8) | 51 (34.0) |
| Blood lactate dehydrogenase increased | 29 (32.6) | 39 (26.0) |
| Anemia | 24 (27.0) | 40 (26.7) |
| Keratitis | 24 (27.0) | 27 (18.0) |
| Hyperuricemia | 21 (23.6) | 36 (24.0) |
| Hypercholesterolemia | 20 (22.5) | 31 (20.7) |
| Dry eye | 19 (21.3) | 36 (24.0) |
| Decreased appetite | 19 (21.3) | 31 (20.7) |
| Hypoalbuminemia | 19 (21.3) | 23 (15.3) |
| Alopecia | 18 (20.2) | 34 (22.7) |
| Amylase increased | 18 (20.2) | 25 (16.7) |
| Hypertriglyceridemia | 18 (20.2) | 25 (16.7) |
| Weight decreased | 16 (18.0) | 28 (18.7) |
| Gamma-glutamyltransferase increased | 15 (16.9) | 28 (18.7) |
| White blood cell count decreased | 13 (14.6) | 27 (18.0) |
| Fatigue | 8 (9.0) | 24 (16.0) |
| CTCAE grade ≥3 study drug-related TEAEs ≥2% for all patients by preferred term | | |
| Hypokalemia | 15 (16.9) | 23 (15.3) |
| Platelet count decreased | 7 (7.9) | 12 (8.0) |
| Neutrophil count decreased | 4 (4.5) | 4 (2.7) |
| Anemia | 3 (3.4) | 7 (4.7) |
| Electrocardiogram QT prolonged | 3 (3.4) | 4 (2.7) |
| Gamma-glutamyltransferase increased | 1 (1.1) | 3 (2.0) |
| Pneumonitis | 1 (1.1) | 3 (2.0) |

AST aspartate aminotransferase, ALT alanine aminotransferase, SAE serious adverse events, TEAEs treatment-emergent adverse events.

were lymph nodes (60.7%), lung (54.0%), bone (42.0%) and liver (42.0%); 11 (7.3%) had brain metastases. Patients with ≥3 metastatic sites accounted for 59.3% (n = 89) across all dose groups. Among HER2-positive patients treated at the RP2D, almost all patients (99%) had visceral metastases. Among all patients enrolled, 88.7% received a median of 3 lines of previous therapy and for those patients enrolled at the RP2D, 93.3% received ≥ to 2 previous lines of therapy. All patients with HER2-positive breast cancer had previously received anti-HER2 therapy, among which the most common treatments were trastuzumab (n = 107, 95.5%), pyrotinib (n = 75, 67.0%) and pertuzumab (n = 38, 33.9%). A small number of patients had also received previous treatment with T-DM1 (n = 6, 5.4%). At the data cutoff date, 97 (64.7%) patients had discontinued treatment, most due to disease progression (n = 77, 79.4%; Fig. 1).

## Safety and tolerability

As the primary end points in phase 1a, dose-limiting toxicities (DLTs) were observed, the maximum tolerated dose (MTD) was not reached, and the recommended phase 2 dose (RP2D) was determined. Two DLTs were observed in the dose-escalation part: one patient experienced a grade 2 decrease in creatinine clearance at 3.0 mg/kg (n = 9), and one patient experienced grade 3 thrombocytopenia with subcutaneous hemorrhage at 3.5 mg/kg (n = 1). The MTD was not reached per protocol definition, and the RP2D was selected to be 2.3 mg/kg

Q3W, based on exposure-response analysis result. The dose regimen of 2.3 mg/kg Q3W was considered to have a better benefit-risk performance with a lower incidence of adverse events (AEs) and without a significant reduction in efficacy when compared to 3.0 mg/kg Q3W[25].

As the secondary end points in phase 1a and 1b, drug-related treatment-emergent AEs (TEAEs) of patients at the RP2D and total are summarized in Table 2 (adverse events of all dose groups are listed in Supplementary Tables 2 and 3).

Overall, drug-related TEAEs were observed in 146 (97.3%) of 150 patients. The most common drug-related TEAEs were aspartate aminotransferase (AST) increased (n = 100, 66.7%), hypokalemia (n = 77, 51.3%) and alanine aminotransferase (ALT) increased (n = 66, 44.0%). Drug-related TEAEs of grade ≥3 were observed in 51 (34.0%) patients, with the most common events being hypokalemia (n = 23, 15.3%) and decreased platelet count (n = 12, 8.0%). Study drug-related serious adverse events (SAEs) were reported in 14 (9.3%) patients, with the most common event being decreased platelet count (n = 4, 2.7%). No left ventricular ejection fraction (LVEF) decreases, nor severe gastrointestinal side effects were observed.

TEAEs led to drug interruption in 46 (30.7%) patients and dose reduction in 37 (24.7%) patients. The most common TEAEs leading to drug interruption or dose reduction were hypokalemia (n = 14, 9.3%), proteinuria (n = 13, 8.7%) and body weight loss (n = 13, 8.7%). TEAEs leading to death were reported in four (2.7%) patients; these were

hemoptysis (*n* = 1, 0.7%), bacterial pneumonia (*n* = 1, 0.7%) and pneumonitis (*n* = 2, 1.3%). Hemoptysis occurred in a patient with an invasive cancer lesion in a main bronchus, and pneumonitis was reported by two patients who had poor baseline lung function, one patient with diffused metastatic lesions in two lungs and one patient who had received radiotherapy for chest lesions several times before.

As noted previously, hypokalemia was a common drug-related TEAE observed in 77 patients (51.3%), with 23 patients (15.3%) experiencing grade ≥3 severity. In all patients with grade ≥3 hypokalemia, blood potassium levels returned to normal or grade 1 after oral or intravenous potassium supplementation. Among 12 patients (8.0%) who experienced grade ≥3 decreased platelet count, no drug discontinuations were needed and no bleeding adverse events were observed except for subcutaneous bleeding in a patient treated in the 3.5 mg/kg group. For this patient, treatment continued after dose reduction and no bleeding happened again. All grade ≥3 events of decreased platelet count recovered after thrombopoietin or platelet receptor agonist treatment. Ocular drug-related TEAEs were observed in 83 (55.3%) patients, the majority of which were grade 1 or 2, and the most frequent ocular toxicities related to the study drug were dry eye (*n* = 36, 24.0%), keratitis (*n* = 27, 18.0%), and dry eye syndrome (*n* = 17, 11.3%). Grade 3 ocular drug-related TEAEs were reported in four (2.7%) patients; two with dry eye, one with blurred vision and one with dry eye syndrome (all receiving 2.3 mg/kg Q3W). All of the ocular TEAEs were reversible, and even the grade 3 toxicity returned to grade ≤1 through the use of supportive measures such as ocular lubricant, topical antibiotic and an anti-inflammatory agent, and other corneal epithelial recovery interventions.

## Antitumor response

Tumor objective response was observed at dose levels of 1.0 mg/kg Q4W and above, with an objective response observed as best response in 48.4% (44/91) of patients with HER2-positive breast cancer across all these dose levels.

Of 67 patients with HER2-positive breast cancer treated at the RP2D of 2.3 mg/kg, two (3.0%) patients had complete response (CR; one to be confirmed) and 34 (50.7%) patients had partial response (PR; seven to be confirmed). The best objective response rate (ORR; confirmed and unconfirmed), the primary end point in phase 1b, was 53.7% (95% confidence interval [CI], 41.1–66.0). The confirmed ORR was 37.5% (95% CI, 25.8-50.0), and the eight unconfirmed patients were confirmed within the next 6 weeks. Secondary efficacy end points are reported as follows. A total of 23 (34.3%) patients experienced stable disease (SD) by data cutoff, which resulted in a disease control rate (DCR) of 88.1% (95% CI, 77.8-94.7) (Table 3 and Fig. 2). Median time to response was 2.7 (95% CI 1.2-not reached) months, and median progression-free survival (PFS) was 15.5 (95% CI, 4.6-not reached) months. Median duration of response (DOR) was not reached, and overall survival (OS) was not mature at data cutoff date. Among 15 efficacy-evaluable patients with HER2-low expression breast cancer treated at the RP2D of 2.3 mg/kg, four (26.7%) patients experienced PR and five (33.3%) patients experienced SD. Of five non-breast cancer patients, two patients, one with lung cancer and one with ampullary cancer, achieved a PR, and three patients experienced SD.

Among the 67 patients with HER2-positive breast cancer treated at the RP2D, an ad hoc, exploratory, retrospective analysis was conducted for those patients who were HR+ (ER+ and/or PR+) and those patients who were HR– (ER– and PR–). The ORRs were 67.6% (23/34, 95% CI, 49.5-82.6) and 40.6% (13/32, 95% CI, 23.7-59.4), respectively. The ORR was 67.9% (19/28, 95% CI, 47.7-84.1), 39.1% (9/23, 95% CI, 19.7-61.5) and 55.6% (15/27, 95% CI, 35.3-74.5) in patients with target lesions in liver, lung and lymph nodes, respectively. In patients with an Eastern Cooperative Oncology Group performance status (ECOG PS) of 0, the ORR was 64.5% (20/31, 95% CI, 45.4-80.8), and in patients with an ECOG PS of 1, the ORR was 44.4% (16/36, 95% CI, 27.9-61.9). In patients aged ≥65 years, the ORR was 37.5% (3/8, 95% CI, 8.5-75.5) and in patients

aged <65 years, the ORR was 55.9% (33/59, 95% CI, 42.4-68.8). ORRs between subgroups were similar regarding factors of previous pertuzumab use, number of previous regimens excluding hormone therapy, and Ki67 index. A forest plot of the subgroup analysis is presented in Supplementary Fig. 1.

## Pharmacokinetics and immunogenicity

Pharmacokinetic analysis of serum concentrations revealed that the exposure of FS-1502 (ADC) increased with increasing dose (Supplementary Fig. 2). At doses above 1.0 mg/kg, the exposure of FS-1502 increased near dose-proportionally with median terminal half-life (t½) ranging from 2.4 days to 4.8 days at cycle 1, while a non-linear trend was observed at dose levels <1.0 mg/kg with shorter half-lives (0.626–2.31 days). Mean clearance of 2.3 mg/kg at cycle 3 was lower than cycle 1 (8.75 mL/day/kg vs 13.0 mL/day/kg), indicating that FS-1502 has time-dependent clearance (Supplementary Table 4). This non-linear PK characteristic is also confirmed in the population PK analysis[25]. There was a slight accumulation after multiple administrations in the Q3W groups, with the accumulation ratio based on the area under the concentration-time curve ranging from 1.04 to 1.71 at 1.0–3.0 mg/kg.

Similar PK profiles were observed for both total antibody and the ADC with concentration and exposure slightly higher for the antibody than those for the FS-1502 (Supplementary Table 5 and Supplementary Fig. 3). The concentration of unconjugated MMAF was generally below 1 ng/mL at each timepoint in each cycle (Supplementary Table 6 and Supplementary Fig. 4), substantially lower than those of the ADC and total antibody (Fig. 3). There was an apparent delay for unconjugated MMAF to reach its maximum serum concentration ($C_{max}$), with a median time to $C_{max}$ ($T_{max}$) of approximately 1–7 days post-infusion.

At the data cutoff date of July 30, 2022, among 81 patients, only five (6.2%) had detectable ADAs, with three (4.2%) patients being ADA positive at baseline. The incidence and magnitude of immunogenicity were low.

## Discussion

This is the first-in-human study of FS-1502, a HER2-targeting ADC with a site-specific conjugated, cleavable tumor-selective linker. FS-1502 was well tolerated, with the majority of TEAEs reported as grade 1 or 2 at the RP2D.

**Table 3 | Antitumor activities of FS-1502 at 2.3 mg/kg (efficacy analysis set)**

|  | 2.3 mg/kg, Q3W |
|---|---|
| Analysis set, *n* | 67 |
| Unconfirmed best response of oncology | |
| CR | 2 (3.0) |
| PR | 34 (50.7) |
| SD | 23 (34.3) |
| PD | 8 (11.9) |
| NE | 0 |
| Best ORR, n (%) | 36 (53.7) |
| 95% CI | 41.1–66.0 |
| DCR, *n* (%) | 59 (88.1) |
| 95% CI | 77.8–94.7 |
| CBR ≥24 weeks, *n* (%) | 40 (59.7) |
| 95% CI | 47.0–71.5 |
| Median PFS, months (95% CI) | 15.5 (4.6-not reached) |

*CBR* clinical benefit rate, *CI* confidence interval, *CR* complete response, *DCR* disease control rate, *NE* not assessable, *ORR* objective response rate, *PD* progressive disease, *PFS* progression-free survival, *PR* partial response, *Q3W* every 3 weeks, *SD* stable disease.
Best ORR: percentage of patients with confirmed and unconfirmed CR and PR.
DCR: the percentage of patients who achieved confirmed CR, PR or SD.
CBR ≥24 weeks: the percentage of patients who achieved confirmed CR, PR or SD for at least 24 weeks.

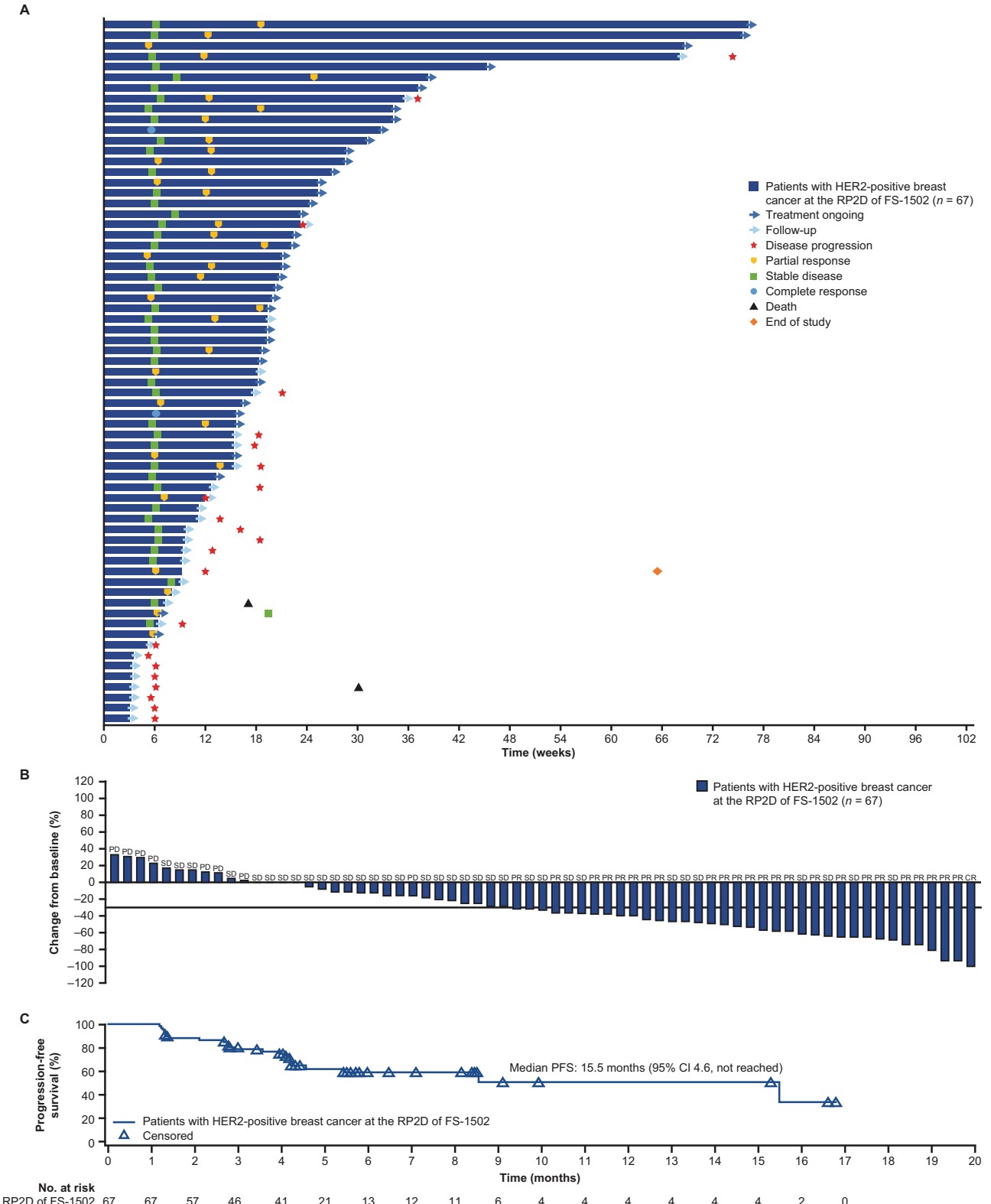

**Fig. 2 | Antitumor activities of FS-1502 at 2.3 mg/kg based on the efficacy analysis set. A** Swimmer plot of treatment duration. **B** Waterfall plot of best percentage change from baseline in total sum of target lesion diameters. **C** Kaplan–Meier curve of progression-free survival. A total of 67 patients were included for the analysis. Progression-free survival is presented as median (95% CI).

Common toxicities reported for other anti-HER2 ADCs are hematological AEs, hepatoxicity, cardiotoxicity, as well as lung and ocular toxicities[26–30]. For FS-1502, low cardiac toxicity (no LVEF decrease and only one asymptomatic G3 QTc prolongation) was observed in the study. Platelet count decrease (mostly grade 1 or 2) was the most common hematologic toxicity in this study, which is also commonly observed with other anti-HER2 ADCs[16,17,29,31–33]. Hypokalemia was reported with the treatment of T-DM1 and disitamab vedotin

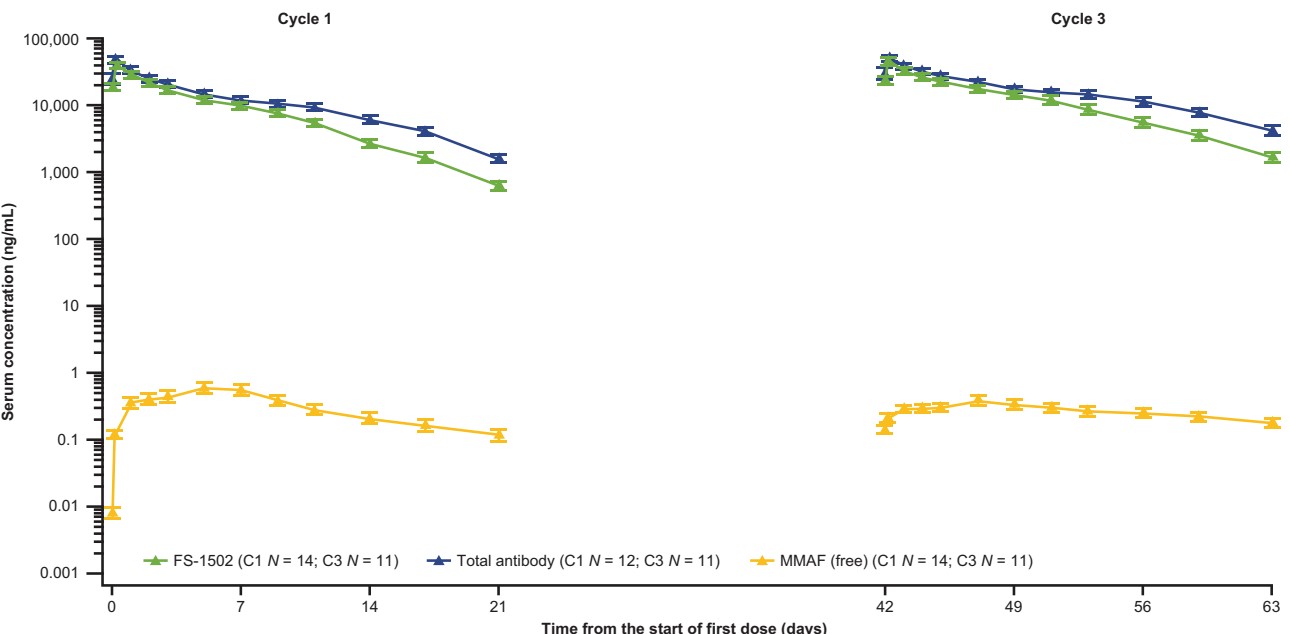

**Fig. 3 | Mean (SD) serum concentration versus time curve of FS-1502, total antibody, and unconjugated MMAF over cycle 1 and cycle 3 at a dose of 2.3 mg/kg.** FS-1502 concentration results of 71 patients were included for cycle 1, and results of 46 patients were included for cycle 3. Data are presented as mean (SD). *C1* cycle 1, *C3* cycle 3, *SD* standard deviation.

(RC48) but with less frequency than seen with FS-1502[26,29]. Low grade ALT and AST increases were observed, suggestive of some hepatoxicity; nothing else was noted in the toxicity results, and all the events were resolved or improved with treatment modifications and/or medical interventions. The incidence rate of ILD/pneumonitis associated with other anti-HER2 ADCs ranged from 8 to 34.8% in previous studies[28,31,33], whereas ILD/pneumonitis did not appear to be a safety signal that needed attention for FS-1502 in the current study. Pneumonitis was observed in five (3.3%) patients, among which two were treated at a high dose level (3.0 mg/kg Q3W) that was not used in further study, while two (2.2%) of 89 patients had pneumonitis at the RP2D. Three (2.0%) experienced grade ≥3 pneumonitis events, two at 3.0 mg/kg Q3W and one (1.1%) at the RP2D. The incidence rate of ILD/ pneumonitis was much lower than other HER2 ADC drugs. Of these five patients, three had poor basal lung function, such as radiation pneumonitis or diffused metastasis in the lungs, which showed the same results with other products.

Ocular toxicity has been noticed in previous studies of ADCs that contain a MMAF payload. Ocular AEs occurred in 77% of patients receiving belantamab mafodotin in a pooled safety analysis, including clinically relevant visual acuity changes in 65% of patients[34]. In the phase 1 study of A166, ocular toxicity was the most common drug-related TEAE, with grade ≥3 events of corneal epitheliopathy (30.9%), blurred vision (18.5%), dry eyes (7.4%) and peripheral sensory neuropathy (6.2%)[35]. In our study, ocular AEs have been managed through sodium hyaluronate artificial eye drops as preventative and continuous administration, as well as close monitoring during the study treatment. Ophthalmologists and investigators treated these AEs according to the principles of corneal epithelial injury management; therapies included sodium hyaluronate artificial eye drops, deproteinized calf blood extract eye drops and recombinant bovine basic fibroblast growth factor eye drops. Ocular AEs were well controlled and recovered under proper intervention and FS-1502 dose modification if needed. The most frequent ocular toxicities related to the study drug were reported as dry eye (24.0%), keratitis (18.0%) and conjunctivitis (14.0%). In a systematic review of ADC-related ocular toxicity, the ocular toxicity seemed to be related to the accumulation of MMAF ADCs within cells, which caused the apoptosis of megakaryocyte progenitors[36].

As of the data cutoff date, objective responses were observed in patients with HER2-postive breast cancer at dose levels of 1.0 mg/kg and higher, and responses were also observed in some patients with HER2-low breast cancer and non-breast cancer patients. The RP2D was determined based on exposure-response analysis, taking into consideration that the toxicity was more frequent and more severe above the 2.3 mg/kg dose level while antitumor efficacy appeared to come to a plateau. Thus, 2.3 mg/kg Q3W was chosen as the optimal regimen with the balance of efficacy and safety in HER2-postive breast cancer. However, it is hard to draw a definitive conclusion given the small sample size. FS-1502 demonstrated encouraging antitumor activity, with the best ORR of 53.7% and median PFS of 15.5 months (95% CI, 4.6-not reached) for patients with HER2-positive breast cancer at the RP2D of 2.3 mg/kg Q3W, with the median DOR and OS not yet reached due to short follow-up. The efficacy of FS-1502 was consistent with that of other ADCs targeting HER2. Phase 3 studies of T-DM1 in patients with HER2-positive breast cancer demonstrated an ORR of 31.0-59.7%[27,30,37]. In a phase 2 study of T-DXd, the response rate (CR plus PR) was 60.9% in patients with previously treated HER2-positive breast cancer[28]. In a phase 3 trial, T-DXd showed a confirmed objective response of 52.3% among all patients with previously treated HER2-low advanced breast cancer[38]. In a phase 1 study of SYD985, the objective response (all PRs) was 33% in patients with HER2-positive breast cancer[31]. In a phase 1 study of ARX 788 in patients with HER2-positive metastatic breast cancer, an ORR of 65.5% was reported[33].

FS-1502 also showed antitumor activity in patients with HER2-low expressing breast cancer, with an ORR of 26.1%, as well as in non-breast cancer patients. However, a definitive conclusion could not be drawn regarding the effectiveness of FS-1502 in these patient populations given the low numbers of patients included in this study. FS-1502 showed higher ORR in patients with HER2-positive than in those with HER2-low breast cancer, it may be related to the hydrophobicity and low bystander cell penetration of the payload effect of MMAF[39]. The ad hoc analysis of this study showed a higher ORR in HR+ patients than HR− patients with HER2-positive breast cancer. The mechanism behind

this is not clear. Crosstalk between the HER2 and HR signaling pathways might be one reason. Estrogen receptors may enhance HER2 signaling activity by promoting the expression of ligands of diverse growth factor receptors[40], which may increase sensitivity to HER2-targeted therapies in HR+ breast cancer. However, longer investigations would be needed to understand the molecular mechanisms behind these observations. Non-breast cancer patients also showed responses to FS-1502 treatment, but any conclusions should be made with caution owing to low numbers of patients, and other further studies are needed, especially for gastrointestinal tumors. In summary, FS-1502 showed its potential in treating HER2-positive breast cancer.

Analysis of FS-1502 pharmacokinetic properties showed a nonlinear pharmacokinetic profile at dose levels <1.0 mg/kg and increased $t_{1/2}$ at higher doses. The reduction in clearance above 1.0 mg/kg is thought to reflect the saturation of HER2-binding sites by the monoclonal antibody, with clearance above 1.0 mg/kg expected to be facilitated by the same physiologic mechanisms that clear other systemic antibodies (typically mediated through binding to Fc receptors). The low concentration of unconjugated MMAF observed suggests FS-1502 is highly stable in plasma following intravenous administration in patients.

There were several limitations in the study. First, this trial was an open-label, single-arm study, and therefore definitive conclusions about efficacy are limited compared to a randomized blinded study. Second, due to safety considerations for a first-in-human trial, the initial dose levels were low, thereby potentially limiting the effectiveness of the treatment in the initial cohorts. Third, HER2 status was retrospective and not confirmed by the central laboratory, therefore the level of HER2 expression at the time of treatment may be different for some patients than projected. Fourth, the study is ongoing, with potential changes in the efficacy and safety results observed after longer follow-up.

In conclusion, FS-1502 was well tolerated and demonstrated promising antitumor activity in patients with HER2-positive advanced breast cancer. Further randomized controlled studies with a larger sample size will be needed to confirm the efficacy and safety of FS-1502 in HER2-positive breast cancer. Phase 2 clinical trials with FS-1502 are currently ongoing in various HER2-positive solid tumor indications, including lung, gastric and colorectal cancer, and a randomized, controlled phase 3 trial comparing FS-1502 with T-DM1 is also being performed in HER2-positive advanced breast cancer.

## Methods
This study adhered to the principles of the Declaration of Helsinki (2013), the International Conference on Harmonisation guidelines for Good Clinical Practice and local applicable regulatory requirements. All patients provided written informed consent before entering the study. Research protocols, amendments and informed consent were approved by relevant institutional review boards and/or ethics committee at nine study sites in China. The ethics committees that approved the study protocol are listed in Supplementary Note 1. The study is registered with ClinicalTrials.gov, NCT03944499.

### Patients
Eligible patients in the dose-escalation part (phase 1a) were adults with HER2-expressing (HER2-high expression: IHC3+, IHC2+/FISH+, or FISH+; HER2-low expression: IHC1+, or IHC2+/FISH-) advanced malignant solid tumors who had failed standard therapy, or could not receive standard therapy, or had no standard therapy available. Patients in the dose-expansion part (phase 1b) had histologically or cytologically confirmed breast cancer with HER2-positive expression, had failed prior anti-HER2 therapy, and had received at least two lines of treatment for advanced breast cancer. The inclusion criteria for

patients in phase 1b were modified in a protocol amendment (version 4.0, Supplementary Note 5), which was approved by the institutional review boards and/or ethics committees. Patients had an ECOG PS of 0 or 1, at least one measurable non-intracranial lesion according to the Response Evaluation Criteria in Solid Tumors (RECIST) version 1.1, adequate organ and bone marrow function, and a life expectancy of ≥12 weeks.

Key exclusion criteria included treatment with anti-HER2 ADCs other than T-DM1 and keratopathy (except for mild punctate keratopathy).

Full details of inclusion and exclusion criteria are provided in the Supplementary Note 2.

### Study design and treatment
This phase 1a/1b, multicenter, open-label, single-arm study was conducted at nine sites in China and consisted of a dose-escalation part and dose-expansion part. A total of 150 patients were enrolled from November 11, 2019, through December 13, 2022. The dose-escalation part evaluated the safety, tolerability, PK and preliminary antitumor activity of FS-1502 in patients with HER2-expressing advanced solid tumors, and the dose-expansion part evaluated the efficacy and safety in patients with locally advanced or metastatic HER2-positive breast cancer. HER2 expression was assessed using immunohistochemistry staining and/or fluorescence in situ hybridization gene amplification testing in tumor tissue sections or fresh tissues[41]. Previous HER2 status reports could be used as the basis for enrollment, and tissue specimens of patients without a HER2 status report prior to enrollment must have been sent to the site for confirmation.

In the dose-escalation phase, patients were enrolled into a traditional 3 + 3 design. Prespecified dose levels included 0.1 (Q4W), 0.2 (Q4W), 0.4 (Q4W), 0.6 (Q4W), 0.8 (Q4W), 1.0 (Q4W and Q3W) and 1.3 (Q4W and Q3W) mg/kg; and the dose-escalation decision was made after comprehensive assessments of PK, pharmacodynamics, safety and efficacy results. If one of three patients in a dose group experienced a DLT, an additional three patients were added to that dose group. If none of the newly added patients out of the total six had a DLT (1/6), the dose would be escalated to the next level. However, if two or more patients out of the total six had a DLT (≥2/6), enrollment in that dose group would be stopped and escalation to the next dose level would not be allowed. The subsequent dose increase ratio was 33% until MTD or RP2D. In the dose-escalation phase, the dose interval initially was 28 d as a cycle but was modified to 21 d as a cycle, starting from the dose of 1.0 mg/kg.

The MTD was determined as the maximum dose of <33% of DLT events observed in patients with evaluable DLT events (i.e., one in six patients at most, or zero in three patients). If the MTD could not be observed, the treatment dose or RP2D would be determined based on the evaluation of antitumor activity, safety and PK data. Details of DLT definition are provided in the Supplementary Note 3.

Based on the antitumor activity, safety and PK data in the dose-escalation study, the safety monitoring committee confirmed the RP2D of FS-1502 treatment to be 2.3 mg/kg Q3W, which was further assessed in the dose-expansion portion of the study. Safety management was performed throughout the study; the study treatment was discontinued and the intervention was given if a patient experienced a grade ≥3 AE. Until the AE returned to grade ≤1, the study treatment was continued with the original or a reduced dose based on the discussion with the investigator.

In the dose-escalation phase, dose adjustments were not allowed during the DLT observation period, but dose interruptions due to non-DLT−related toxicities were allowed during the first cycle. Dose interruption was allowed after the DLT observation period in the dose-escalation phase and dose-expansion phase. Details of dose modification rules are provided in the Supplementary Note 4.

## End points

The primary end points of the dose-escalation part were DLTs (defined as predefined toxicities that occur during the DLT observation period, details are provided in the Supplementary Note 3), the MTD (defined as the maximum dose of <33% of DLT events observed in patients with evaluable DLT events) and the RP2D (determined based on analysis of PK/PD, safety and efficacy results). Secondary end points included safety end points other than DLTs, such as incidence of TEAEs, SAEs, and TEAEs leading to drug discontinuation/death. Other secondary end points were ORR (defined as the proportion of patients with confirmed CR and PR according to RECIST version 1.1) assessed by the investigator, PFS (defined as the time from the first dose of study treatment to disease progression or death, whichever occurred first), OS (defined as the time from the initiation of study treatment to death due to any cause), 1-year OS rate (defined as the proportion of patients that survived within 1 year of the initiation of study treatment), DOR (defined as the time from first CR or PR to disease progression or death due to any cause, whichever occurred first) clinical benefit rate (CBR; defined as the proportion of patients with CR, PR and SD lasting ≥24 weeks according to RECIST version 1.1) and PK parameters of FS-1502, total antibody and unconjugated MMAF (including maximum concentration, half-lives, area under the serum concentration-time curves, clearance and accumulation ratio, etc), the ADA and the NAb of FS-1502. DCR (defined as the proportion of patients with CR, PR and SD lasting ≥6 weeks) based on the investigator's assessment was not a prespecified end point, but was also analyzed in the efficacy analysis.

The primary end point of the dose-expansion part was to evaluate the IRC-assessed ORR of patients with HER2-positive breast cancer. Secondary end points included safety (such as incidence of TEAEs, SAEs, TEAEs leading to drug discontinuation, and frequency and cause of death within 30 days after the last dose), PFS, OS, 1-year OS rate, DOR, CBR and PK parameters of FS-1502, total antibody and unconjugated MMAF, the ADA and the NAb of FS-1502. DCR, which was not a prespecified end point, was also analyzed in the efficacy analysis.

## Definition of study end points

- DLT: Toxicities occurring during the DLT observation period (Day 21/Day 28 after the first dosing).
- MTD: Maximum dose of <33% DLT events observed in patients with evaluable DLT events (i.e., one in six patients at most or zero in three patients).
- ORR: Proportion of patients with CR and PR according to RECIST version 1.1.
- PFS: Time from the first dose of study treatment to disease progression or death, whichever occurred first. The cutoff date for patients without disease progression or death was the last tumor assessment; for patients without post-baseline tumor assessments, it was the first dose administration.
- OS: Time from the initiation of study treatment to death from any cause. The cutoff date for patients who did not die was the date the last patient survival information was obtained.
- DOR: Time from first CR or PR to disease progression or death from any cause, whichever occurred first. In the absence of tumor progression or death, the cutoff date was the date of the last tumor assessment.
- 1-year OS rate: Proportion of patients who survived within 1 year of the initiation of study treatment.
- DCR: Proportion of patients with CR, PR and SD lasting ≥6 weeks.
- CBR: Proportion of patients with CR, PR and SD lasting ≥24 weeks according to RECIST version 1.1.

## Study assessments

The safety assessments of FS-1502 included TEAEs, SAEs, laboratory tests, electrocardiogram, and vital signs; graded per the National Cancer Institute Common Terminology Criteria for Adverse Events version 5.0. AEs were recorded throughout the trial until 30 d after the last dose administration or the initiation of new antitumor therapy, whichever occurred first. The causality of a TEAE to the study drug was prespecified: it was considered not drug related if no study drug was used or the time of the AE was not related to the use of study drug or the cause of the AE was otherwise clear, and considered drug related if there was evidence of the use of study drug and the occurrence of the TEAE was reasonably related in time to the use of study drug. Ocular toxicity was monitored during the study; study drug modifications and interventions were taken per the prespecified management plan. The management plan for ocular toxicity was added in a protocol amendment (version 6.0, Supplementary Note 5), which was approved the institutional review boards and/or ethics committees.

The efficacy of FS-1502 was evaluated by radiographic tumor assessment (computed tomography or magnetic resonance imaging) as per RECIST version 1.1. Radiographic tumor assessment was performed every two cycles.

Blood samples were collected for the measurements of the systemic levels of FS-1502, unconjugated MMAF, total antibody and other parameters. Serum concentrations of FS-1502 and total antibody were determined by a validated sandwich enzyme-linked immunosorbent assay; the lower limit of quantitation of FS-1502 and total antibody was 50 ng/mL. Serum concentrations of unconjugated MMAF were determined by a liquid chromatography-tandem mass spectrometry assay with the lower limit of quantitation of 4.75 pg/mL. Pre-dose samples were collected at all treatment cycles, and intensive blood sample collection after dosing was performed at treatment cycle 1 and 3.

The immunogenicity of FS-1502 was assessed with the levels of ADA and NAb. ADA in serum was detected with an electro-chemiluminescence assay; all confirmed ADA-positive samples were further assessed for the presence of NAb. Pre-dose blood samples were collected for ADA measurement until the last treatment cycle.

## Statistical analysis

The dose-escalation study enrolled patients with a 3 + 3 approach, and approximately 92 patients were estimated for enrollment. Approximately 50 patients were estimated for enrollment in the dose-expansion part, with the assumption that the ORR of FS-1502 was 50%, type I error was 0.025 for single side, and the 95% CI was 35.5-65.4%. In a protocol amendment (version 6.0), which was approved by the institutional review boards and/or ethics committees, the estimated number of patients for enrollment was adjusted to 200 based on study enrollment status and subsequent development strategy; the actual number of patients enrolled was 150 due to changes in the research and development strategy.

A protocol amendment (version 6), approved the institutional review boards and/or ethics committees, specified that an interim analysis would be conducted once a certain amount of data had accumulated as assessed using the frequentist method. Interim analysis was conducted after data from 65 patients were collected in the phase 1b trial. The sponsor comprehensively considered safety and effectiveness, as well as the competing product data, to make decisions.

DLT was analyzed in the DLT set (defined as patients with DLTs that occurred during the DLT observation period) and patients without DLTs during the DLT observation period who had taken at least 80% of the planned dose amount. Safety was analyzed in the safety analysis set, including any patient who received at least one dose of FS-1502 and a safety evaluation. Efficacy was assessed in the efficacy analysis set, consisting of patients who received at least one dose of FS-1502 and had at least one baseline and one post-baseline assessment of tumor response. Serum concentrations of FS-1502, MMAF and total antibody were analyzed in the PK concentration analysis set, defined as patients who received at least one dose of FS-1502, with at least one

blood sample, and had FS-1502 concentration data. PK parameters were assessed in the PK parameter analysis set, defined as patients who received FS-1502 per protocol with at least one PK parameter data element available during the study. Patients with a protocol violation, PK parameter results that were affected or the parameters could not be estimated were excluded. Immunogenicity was tested in the immunogenicity analysis set, defined as patients who had taken at least one dose of FS-1502, with at least one blood sample, and had the FS-1502 antibody concentration data.

Efficacy and safety data from the dose-escalation and dose-expansion parts were pooled for analysis. The ORR 95% Clopper–Pearson CIs were performed. PFS, OS and DOR were analyzed by Kaplan–Meier curves, and descriptive statistics of 1-year OS rate, DCR and CBR were evaluated according to independent review committee and investigator assessments.

All data were collected via Medidata Classic Rave System (Version: Medidata Classic Rave® 2023.1.1). Statistical analysis was performed using SAS version 9.4, and PK data analysis was performed using WinNonlin software.

### Reporting summary

Further information on research design is available in the Nature Portfolio Reporting Summary linked to this article.

## Data availability

The de-identified participant data that support the findings of this study are available from the corresponding author by contacting xubinghe@medmail.com.cn for research purposes and will be responded to within 4 weeks. Source data are provided with this paper. The study protocol can be found in the Supplementary Information as Supplementary Note 5. All remaining data can be found in the Article, Supplementary and Source data files. Source data are provided with this paper.

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

## Acknowledgements

This study was funded by Shanghai Fosun Pharmaceutical Industrial Development Co., Ltd., and partly supported by the Chinese Academy of Medical Sciences Innovation Fund for Medical Sciences (CIFMS: 2021-I2M-1-014 and 2022-I2M-2-002, received by B. X.). Medical writing and editorial support were provided by JingYi Lee, PhD, of Parexel and funded by Shanghai Fosun Pharmaceutical Industrial Development Co., Ltd.. Shanghai Fosun Pharmaceutical Industrial Development Co., Ltd., participated in the study design, data collection and analysis and manuscript writing.

## Author contributions

Study conception and design: B.X., Q.L., Zhuli W., A.H. Project supervision: B.X., Q.L., Zhuli W., Y.J., L.D., Y.S., Y.Z., Xingli W. Participant recruitment and coordination: B.X., Q.L., Y.C., Z.T., Y.L., Xian W., M.Y., J.C., S.W., C.D., L.L., C.W., L.L., C.W., M.W., Zhuo W. Data collection and processing: Q.L., Y.C., Z.T., Y.L., X.W., M.Y., J.C., S.W., C.D., L.L., C.W., L.L., C.W., M.W., Zhuo W., Zhuli W., Y.J., L.D., Y.S., Y.Z. Manuscript preparation: Q.L., Zhuli W., Y.J., L.D., Y.S., Y.Z. Manuscript review and editing: all co-authors.

## Competing interests

Zhuli W., Xingli W., Y.J., L.D., Y.S. and Y.Z. are employees of Fosun Pharmaceutical Development Co., Ltd.. B.X. reports receiving advisory fees from Novartis and AstraZeneca and fees for serving on a speakers' bureau from Pfizer and Roche. Other authors have no competing interests.
