## [Peer Review File · Nature Communications]

Reviewers' Comments:

Reviewer #1:

Remarks to the Author:

Thank you for the opportunity to review the manuscript entitled "Phase 1a/b study 1 of FS-1502, a Novel anti-HER2 Antibody-Drug Conjugate, in Patients with 2 HER2-Positive Metastatic Breast Cancer". The manuscript addresses a timely topic, as antibody-drug conjugates have emerged as a powerful strategy in cancer therapy. Specifically, this trial enrolled 150 patients, of whom 145 were diagnosed with metastatic breast cancer. There were two dose-limiting toxicities and the maximum tolerated dose (MTD) was not determined, and the recommended phase 2 dose (RP2D) was found to be 2.3 mg/kg every 3 weeks, with an objective response rate (ORR) of 53.7% (unconfirmed) and a median progression-free survival (mPFS) of 15.5 months in patients with HER2+ breast cancer who received the RP2D (n = 67).

Some comments/recommendations:

1. Consider changing the title concerning the definition of 'HER2+ metastatic breast cancer' to 'HER2-expressing' since HER2-low tumors were included.
2. Include the registration number for this trial (NCT03944499).
3. In the methods section, please include the formal characteristics of the clinical trial.
 - a. Mention that this was an open-label, single-arm, dose-escalation, and dose-expansion phase 1 trial.
 - b. Clarify that, while other solid tumors were allowed for the dose escalation part, only patients with metastatic HER2+ breast cancer were eligible for the dose-expansion part.
 - c. Also, mention the number of sites in China involved, as it was a multicenter clinical trial. Clearly state the primary endpoints of safety, such as identifying DLTs, MTD, and RP2D for patients who received at least one dose of the study drug in the dose-escalation phase. Specify the primary endpoint for the dose expansion part (ORR)—was it prespecified to be confirmed responses?— if confirmed responses was specified, the ORR should be updated in the abstract. In the results section of the abstract, provide a summary of key findings, including the accrual period, study cutoff date for safety data analysis, and median follow-up. Highlight the main safety observations mentioned in the original version of the abstract.
4. In the introduction, it is necessary to include references and information to place this clinical trial in the current clinical context regarding the management of HER2-positive breast cancer, particularly considering the current clinical use of other ADCs such as T-DM1 and T-DXd.
5. Need more information about the main rules for assessing safety and determining the RP2D. The 'End points' paragraph also needs restructuring as it currently presents only a list of outcomes. Include a brief description of the RECIST version used for radiologic assessment. Specify the main rules for early study discontinuation, noting that an interim analysis using Bayesian posterior probability was conducted when approximately 20 patients had completed two tumor assessments. If the predicted Pr (ORR <20%) was >80%, indicating that fewer than 6 responses were observed in 20 evaluable patients, there was an 80% probability that the drug ORR was lower than standard of care. In the 'study procedure' section, clarify the rules regarding dose interruptions and dose reductions (whether they were allowed or not and how they were carried out), as per the study protocol. Include rules for further escalation after prior toxicity, specifying whether it was allowed or not and how it was managed. In the 'study procedure' section, detail how HER2 status was assessed and whether central laboratory confirmation was required, as described in the study protocol for both the escalation and the expansion phases.
6. Need to have a full AE table that has grading separated g1-4 so we can better understand the toxicity profile

Reviewer #2:

Remarks to the Author:

This trial has investigated the safety and efficacy of FS-1502 in patients with HER2-expressing

solid tumors and HER2-positive breast cancer. The overall workflow of the study design is relatively reasonable, and the major conclusions are supported by the study data.

However, there still exist some minor points to address.

1. Please check the Figure 1 carefully for the accurate number, in which one row (1.3mg/kg Q4w) do not correspond with each other.

2. Six patients had received previous treatment with T-DM-1, what about the efficacy?

3. Ad hoc, exploratory, retrospective analysis demonstrated the ORRs were 67.6% (23/34, 95% CI, 49.5-82.6) and 40.6% (13/32, 95% CI, 23.7-59.4) in patients with HR positive subgroup and HR negative subgroup, respectively. Please explain the efficacy disparity in the Discussion.

4. T-DXd is a potent ADC, which exhibits promising profile in patients with HER2 positive and HER2 low breast cancer. The efficacy and safety comparison with T-DXd should be supplemented.

Reviewer #3:

Remarks to the Author:

Many thanks for the opportunity to review this manuscript, it is a neat explanation of the observations from this study.

I have the following comments:

- Abstract:

line 40: the number of decimal places doesn't match with confidence intervals for the two different outcomes here. Please match estimates and their confidence intervals.

Figure 2C: why does the number at risk stop after only a few time points?

Responses to reviewers' comments

Dear reviewer,

On behalf of my co-authors, I would like to thank you for considering our manuscript “Phase 1a/b study of FS-1502, a Novel anti-HER2 Antibody-Drug Conjugate, in Patients with HER2-Positive Metastatic Breast Cancer” (Manuscript # NCOMMS-23-44187-T) for publication in *Nature Communications*. We have revised the manuscript and have provided a point-by-point response to the reviewers' comments below.

No.	Editors'/reviewers' comments	Author response and changes made	Line number in the tracked version	Line number in the clean version
Reviewer 1				
1	Consider changing the title concerning the definition of 'HER2+ metastatic breast cancer' to 'HER2-expressing' since HER2-low tumors were included.	We appreciated the reviewer's recommendation. The title has been revised as “Phase 1a/b study of FS-1502, a Novel anti-HER2 Antibody-Drug Conjugate, in Patients with HER2-Expressing Metastatic Breast Cancer”	Main text: line 2	Main text: line 2
2	Include the registration number for this trial (NCT03944499).	The registration number is added in line 34 the abstract, and also already included in line 348, the last sentence of the section “Study design and treatment”. Line 34, This multicenter, open-label, single-arm, dose-escalation and dose-expansion phase 1 trial (NCT03944499) was conducted at nine sites in China. Line 348, The study is registered with ClinicalTrials.gov, NCT03944499.	Main text: line 34; line 348	Main text: line 34; line 344
3	In the methods section, please include the formal characteristics of the clinical trial. a. Mention that this was an open-label, single-arm, dose-escalation, and dose-expansion phase 1 trial. b. Clarify that, while other solid tumors were allowed for the dose escalation part, only patients with metastatic HER2+ breast cancer were eligible for the dose-expansion part. c. Also, mention the number of sites in	We have added the requested information in the methods section of the abstract. a. In lines 33-35, we added: This multicenter, open-label, single-arm, dose-escalation and dose-expansion phase 1 trial (NCT03944499) was conducted at nine sites in China. b. In lines 35-37, we added: Patients with HER2-expressing advanced solid tumors were eligible for the dose-escalation part, and patients with metastatic HER2-positive breast cancer (BC) were eligible for the dose-expansion part. c. In lines 33-35, we added: This multicenter , open-label, single-arm, dose-escalation and dose-expansion phase 1	Main text: lines 33-45	Main text: lines 33-43

	China involved, as it was a multicenter clinical trial. Clearly state the primary endpoints of safety, such as identifying DLTs, MTD, and RP2D for patients who received at least one dose of the study drug in the dose-escalation phase. Specify the primary endpoint for the dose expansion part (ORR)—was it prespecified to be confirmed responses?-- if confirmed responses was specified, the ORR should be updated in the abstract. In the results section of the abstract, provide a summary of key findings, including the accrual period, study cutoff date for safety data analysis, and median follow-up. Highlight the main safety observations mentioned in the original version of the abstract.	trial (NCT03944499) was conducted at nine sites in China. For primary endpoints of safety, we added in lines 37-39: The primary end points were dose-limiting toxicities (DLTs), maximum tolerated dose (MTD) and recommended phase 2 dose (RP2D) for the dose-escalation part and objective response rate (ORR) for the dose-expansion part. The primary endpoint for the dose expansion part (ORR) was prespecified to be confirmed responses. In lines 37-39, we added: The primary end points were dose-limiting toxicities (DLTs), maximum tolerated dose (MTD) and recommended phase 2 dose (RP2D) for the dose-escalation part and objective response rate (ORR) for the dose-expansion part. In lines 39-41, the accrual period of enrollment was added: From November 11, 2019, to December 13, 2022, 150 patients with HER2-expressing solid tumors ($n = 5$) and BC ($n = 145$) were enrolled in phase 1a and 1b, respectively. In lines 42-43, the study cutoff date for safety data analysis was added: As of the data cutoff date (December 24, 2022), one DLT each was reported at 3.0 and 3.5 mg/kg. In lines 44-45, median follow-up was added: The median duration of follow-up was 5.8 months. In lines 42-44, the main safety observations mentioned in the original version of the abstract were added back: As of the data cutoff date (December 24, 2022), one DLT each was reported at 3.0 and 3.5 mg/kg. The MTD was not defined; an RP2D of 2.3 mg/kg once every 3 weeks was selected.		
4	In the introduction, it is necessary to include references and information to place this clinical trial in the current clinical context	We appreciated the reviewer’s recommendation. In the introduction, lines 66-69, we added references and a description of the current clinical studies regarding the	Main text: lines 66-69	Main text: lines 64-67

	regarding the management of HER2-positive breast cancer, particularly considering the current clinical use of other ADCs such as T-DM1 and T-DXd.	management of HER2-positive breast cancer: T-DXd is recommended as the second-line treatment for the management of HER2-positive and HER2-low breast cancers, and T-DM1 is recommended as the third-line treatment for patients with metastatic HER2-positive breast cancer¹⁶.		
5	Need more information about the main rules for assessing safety and determining the RP2D. The 'End points' paragraph also needs restructuring as it currently presents only a list of outcomes. Include a brief description of the RECIST version used for radiologic assessment. Specify the main rules for early study discontinuation, noting that an interim analysis using Bayesian posterior probability was conducted when approximately 20 patients had completed two tumor assessments. If the predicted Pr (ORR <20%) was >80%, indicating that fewer than 6 responses were observed in 20 evaluable patients, there was an 80% probability that the drug ORR was lower than standard of care. In the 'study procedure' section, clarify the rules regarding dose interruptions and dose reductions (whether they were allowed or not and how they were carried out), as per the study protocol. Include rules for further escalation after prior toxicity, specifying whether it was allowed or not and how it was managed. In the 'study procedure' section, detail how HER2 status was assessed and whether central laboratory confirmation was required, as described in the study protocol for both the escalation and the expansion phases.	In lines 350-353, more information about the main rules for assessing safety and determining the RP2D was added: The primary end points of the dose-escalation part were DLTs (defined as predefined toxicities that occur during the DLT observation period, details are provided in the Supplementary Appendix), MTD (defined as the maximum dose of <33% of DLT events observed in patients with evaluable DLT events) and RP2D (determined based on analysis of PK/PD, safety and efficacy results). In lines 350-366, The 'End points' paragraph was restructured with end points definitions: The primary end points of the dose-escalation part were DLTs (defined as predefined toxicities that occur during the DLT observation period, details are provided in the Supplementary Appendix), MTD (defined as the maximum dose of <33% of DLT events observed in patients with evaluable DLT events) and RP2D (determined based on analysis of PK/PD, safety and efficacy results). Secondary end points included safety end points other than DLTs, such as incidence of TEAEs, TEAEs leading to drug interruption/reduction and the frequency and cause of death. Other efficacy end points of were ORR (defined as the proportion of patients with confirmed CR and PR according to RECIST version 1.1) assessed by the investigator, PFS (defined as the time from the first dose of study treatment to disease progression or death, whichever occurred first), OS (defined as the time from the initiation of study treatment to death due to any cause), 1-yr OS rate (defined as the proportion of patients that survived within 1 year of the initiation of study treatment), DOR (defined as the time from first CR or PR to disease progression or death due	Main text: lines 350-366; lines 398-400; lines 339-343; lines 321-325; lines 313-317 Supplement information: Dose modification rules	Main text: lines 346-362; lines 394-396; lines 335-339; lines 318-322; lines 310-314 Supplement information: Dose modification rules

		to any cause, whichever occurred first), DCR (defined as the proportion of patients with CR, PR and SD lasting \geq 6 weeks), and clinical benefit response (CBR, defined as the proportion of patients with CR, PR and SD lasting \geq 24 weeks according to RECIST version 1.1), PK parameters of FS-1502 (including maximum concentration, half-lives, area under the serum concentration-time curves, clearance and accumulation ratio, etc), total antibody and unconjugated MMAF, ADA and the neutralizing antibody (NAB) of FS-1502. In lines 355-357, a brief description of the RECIST version used for radiologic assessment was included: Other efficacy end points were ORR (defined as the proportion of patients with confirmed CR and PR according to RECIST version 1.1) assessed by the investigator... In lines 398-400 of the “Statistical analysis” section, the main rules for early study discontinuation were specified: An interim analysis using the frequentist method was conducted after a certain amount of data was collected in the trial. The sponsor comprehensively considered safety and effectiveness as well as the competing product data to make decisions. In lines 339-343 of the “Study design and treatment” section, the rules regarding dose interruptions and dose reductions (whether they were allowed or not and how they were carried out) as per the study protocol were clarified: In the dose-escalation phase, dose adjustments were not allowed during the DLT observation period, but dose interruptions due to non-DLT-related toxicities were allowed during the first cycle. Dose interruption was allowed after the DLT observation period in the dose-escalation phase and dose-expansion phase. Details of dose modification rules are provided in the Supplementary Appendix. In lines 321-325 of the “Study design and treatment” section, rules for further escalation after prior toxicity was specified:		
--	--	---	--	--

		If one of three patients in a dose group experienced a DLT, an additional three patients were added to that dose group. If none of the newly added patients out of the total six had a DLT (1/6), the dose would be escalated to the next level. However, if two or more patients out of the total six had a DLT ($\geq 2/6$), enrollment in that dose group would be stopped and escalation to the next dose level would not be allowed. In lines 313-317 of the “Study design and treatment” section, HER2 status assessment per protocol was described: HER2 expression was assessed using immunohistochemistry staining and/or fluorescence in situ hybridization gene amplification testing in tumor tissue sections or fresh tissues. Previous HER2 status reports could be used as the basis for enrollment, and tissue specimens of patients without a HER2 status report prior to enrollment must be sent to the site for confirmation.		
6	Need to have a full AE table that has grading separated g1-4 so we can better understand the toxicity profile	We have provided a full AE table that has grading separated g1-4 as Supplementary Table 3. Supplementary Table 3 is also listed in lines 121-122: Adverse events of patients at RP2D and total are summarized in Table 2 (adverse events of all dose groups are listed in Supplementary Table 2 and 3).	Supplement information: Supplementary Table 3 Main text: lines 121-122	Supplement information: Supplementary Table 3 Main text: lines 118-119
	Reviewer 2			
1	Please check the Figure 1 carefully for the accurate number, in which one row (1.3mg/kg Q4w) do not correspond with each other.	We appreciated the reviewer’s reminding. The number of termination reasons has been corrected as seen in revised Figure 1.	Main text: Page 24 Figure 1	Main text: Page 24 Figure 1
2	Six patients had received previous treatment with T-DM-1, what about the efficacy?	The efficacy of the six patients who had received prior T-DM-1 were: 4 patients with 2 SD and 2 PD in the dose group of 2.3 mg/kg; 1 patient with PD in the dose group of 0.4 mg/kg; and 1 patient in the dose group of 1.0 mg/kg withdrew ICF before the first tumor assessment, therefore, the efficacy result was not available.	Not applicable	Not applicable
3	Ad hoc, exploratory, retrospective analysis demonstrated the ORRs were 67.6% (23/34, 95% CI, 49.5-82.6) and 40.6% (13/32, 95%	In lines 263-269 of the “Discussion” section, we tried to explain the efficacy imparity: The ad hoc analysis of this study showed a higher ORR in HR+ patients than HR-	Main text: lines 263-269	Main text: lines 260-266

	CI, 23.7-59.4) in patients with HR positive subgroup and HR negative subgroup, respectively. Please explain the efficacy imparity in the Discussion.	patients with HER2-positive breast cancer. The mechanism behind this is not clear. Crosstalk between the HER2 and HR signaling pathways might be one reason. Estrogen receptors may enhance HER2 signaling activity by promoting the expression of ligands of diverse growth factor receptors⁴¹, which may increase sensitivity to HER2-targeted therapies in HR+ breast cancer. However, longer investigations would be needed to understand the molecular mechanisms behind these observations.		
4	T-DXd is a potent ADC, which exhibits promising profile in patients with HER2 positive and HER2 low breast cancer. The efficacy and safety comparison with T-DXd should be supplemented.	In lines 216-218, the safety of FS-1502 and other HER2 ADCs (including T-DXd) was already compared: The incidence rate of ILD/pneumonitis associated with other anti-HER2 ADCs ranged from 8 to 34.8% in previous studies^{28,33,35}, whereas ILD/pneumonitis did not appear to be a safety signal that needed attention for FS-1502 in the current study. In lines 252-255, the efficacy of FS-1502 and T-DXd was already compared: In a phase 2 study of T-DXd, the response rate (complete response plus partial response) was 60.9% in patients with previously treated HER2-positive breast cancer²⁷. In a phase 3 trial, T-DXd showed a confirmed objective response of 52.3% among all patients with previously treated HER2-low advanced breast cancer³⁹.	Main text: lines 216-218, lines 252-255	Main text: lines 213-215, lines 249-252
Reviewer 3				
1	line 40: the number of decimal places doesn't match with confidence intervals for the two different outcomes here. Please match estimates and their confidence intervals.	Thank you for reminding. We have revised the data values in lines 46-50: Of 67 HER2-positive BC patients receiving the RP2D, the best ORR was 53.7% (95% CI, 41.1-66.0%), including PR confirmed (the confirmed ORR was 37.5%) and pending for confirmation, and median progression-free survival was 15.5 months (95% CI, 4.6-not reached).	Main text: lines 46-50	Main text: lines 45-47
2	Figure 2C: why does the number at risk stop after only a few time points?	An updated figure 2C was provided. The number at risk was 13 at month 6, 6 at month 9, 4 at month 12 and 15, and 0 at month 18.	Main text: page 26 Figure 2	Main text: page 26 Figure 2

Reviewers' Comments:

Reviewer #1:

Remarks to the Author:

This is a phase 1a/b (dose escalation and dose expansion) multi-center clinical trial. The trial enrolled patients with HER2-expressing solid tumors (n=5) and specifically HER2-positive metastatic breast cancer (n = 145), in the dose escalation and dose expansion parts, respectively. The primary endpoints of the study were Dose-Limiting Toxicities (DLTs) for the dose-escalation part and Overall Response Rate (ORR) for the dose-expansion part. At the data cutoff, 1 DLT each was reported at 3.0 and 3.5 mg/kg, with no MTD defined, and a RP2D of 2.3 mg/kg once every 3 weeks selected.

The title and abstract sections have been revised according to the reviewers' suggestions. Specifically, the trial registration number has been added, as well as more detailed safety information (e.g., pneumonitis). In addition to best ORR, the authors added confirmed Partial Response (PR). The title now states "HER2-expressing" tumors, which better aligns with a trial that included both a dose escalation and a dose expansion phase, with other tumors allowed in the former phase, in addition to the fact that the trial involved both HER2-positive and HER2-low tumors.

In the introduction, I appreciate the authors for adding concise introductory information about antibody-drug conjugates as a drug class, with a brief description of their structure. A brief description of the positioning of currently used HER2-directed ADCs in the management of patients with metastatic breast cancer has also been added as required. I would suggest a more precise wording regarding the clinical indication for T-DXd and T-DM1. T-DXd has been approved in unresectable/metastatic HER2-positive breast cancer who received a prior anti-HER2-based regimen either in the metastatic setting or in the (neo)adjuvant setting and have developed disease recurrence during or within six months of completing therapy; it has been also approved in unresectable/metastatic HER2-low (i.e., IHC 1+ or 2+/ISH-) breast cancer who have received a prior chemotherapy in the metastatic setting or developed disease recurrence during or within 6 months of completing adjuvant chemotherapy. I suggest also improving the indication for T-DM1. Indeed, after the positive results in terms of median Progression-Free Survival (mPFS) and Overall Survival (OS) of HER2CLIMB (tucatinib-capecitabine-trastuzumab) and DESTINY-Breast03 (T-DXd), T-DM1 is now recommended in the metastatic setting after progression on either/both regimens by international medical oncology guidelines.

I appreciate the authors addressing the major concerns related to the method section. The patient population paragraph has been enriched with information about eligibility criteria for each cohort. The authors improved the description of the formal characteristics of this clinical trial. Specifically, they noted that this is a multicenter, open-label, single-arm, dose-escalation and dose-expansion phase 1 trial (NCT03944499) conducted at nine sites in China (section: Study design and treatment). The authors clarified that solid tumors were allowed for the dose escalation part, while only patients with metastatic HER2+ breast cancer were eligible for the dose-expansion part. The primary endpoints for each phase of the study (dose escalation and dose expansion) have been better explained, as well as the study cutoff date for safety data analysis and median follow-up. In the study design section, the rules for dose escalation have been specified.

The Study design section has been enriched with information regarding the main rules for assessing safety and determining the RP2D, as required. Specifically, if one of three patients in a dose group experienced a DLT, an additional three patients were added to that dose group (3+3 design). If none of the newly added patients out of the total six had a DLT (1/6), the dose would be escalated to the next level. However, if two or more patients out of the total six had a DLT ($\geq 2/6$), enrollment in that dose group would be stopped, and escalation to the next dose level would not be allowed. The subsequent dose increase ratio was 33% until MTD or RP2D. Dose modification rules have been added to the supplementary files. HER2 assessment has been described in the revised version.

The 'End points' section has been restructured. Instead of a previous list of endpoints, the authors now provide a concise definition in brackets for each primary safety endpoint of the dose-

escalation phase, citing the supplementary appendix for full information (e.g., pages 7-8, marked version).

However, in Phase Ia, I suggest describing the endpoints as reported in the attached study protocol. For example, in line 352, ORR is described as "other efficacy endpoints". In the protocol, the incidence of Dose-Limiting Toxicities (DLTs) is the primary safety endpoint, and ORR is the primary efficacy endpoint. Other efficacy endpoints were only Progression-Free Survival (PFS), Overall Survival (OS), 1-year OS, Duration of Response (DOR), and Clinical Benefit Rate (CBR) (see page 2143 of the study protocol). Disease Control Rate (DCR) was not pre-specified in the protocol. Endpoint abbreviations should be correctly explained when they first appear in the text. Some of those like ORR, DLT, MTD, RP2D have been described in the abstract. However, CBR should be clinical benefit rate (not response, see line 358). Also, I would suggest making the nomenclature of the secondary safety endpoints (e.g., "any AEs" incidence, "TEAEs", "drug-related AEs", "drug-related TEAEs") more consistent (see protocol versus supplementary tables versus revised version of the manuscript).

In Phase Ib, one of the secondary endpoints included in the method section (lines 364-366) is safety. Please, concisely add what safety means here, therefore frequency of AEs, grade and SAEs leading to treatment discontinuation. Specify if a correlation with drug exposure was pre-specified in the protocol [i.e., AEs "drug-related" versus "not-drug-related"] or not [i.e., any AEs since investigational drug started]), then improve nomenclature consistency of AEs as above. Other secondary endpoints were PFS, OS, 1-year OS rate, DOR, CBR, PK parameters of FS-1502, total antibody and unconjugated MMAF, the ADA and the NAB of FS-1502. Please add that another secondary endpoint was frequency and cause of death within 30 days after the last dose. Specify in the method section that DCR was not a pre-specified endpoint in the protocol (see page 2143); then explain that you additionally report that in the results.

RECIST version used has been reported now ('end points' and 'study assessments' paragraphs).

In the statistical analysis section (line 394), we find: "An interim analysis using the frequentist method was conducted after a certain amount of data was collected in the trial." This sentence looks a bit ambiguous as it is not clear when a "certain amount of data" was collected. The protocol reports that "an interim analysis using Bayesian posterior probability will be conducted when approximately 20 patients have completed 2 tumor assessments. If the predicted Pr (ORR <20%) is >80%, i.e. fewer than 6 responses are observed in 20 evaluable patients, then there is an 80% probability that the drug ORR is lower than standard of care and early discontinuation of the cohort may be considered, otherwise continuing enrollment to approximately 50 patients. Bounds will be adjusted based on the actual number of people in the Efficacy Analysis Set. The interim analysis margins are non-binding and the sponsor will consider the safety and effectiveness data in the final decision." Moreover, if the highlighted sentence above was added to better specify the main rules for study discontinuation as mentioned in the rebuttal letter, I believe the goal of early discontinuation and pre-specified rules for the analysis should be better explained in the text, as well.

In the results section, I would recommend using the wording "patients with breast cancer", instead of breast cancer patients, as it is more inclusive of the instances of patient advocates.

In Supplementary files a new table summarizing Treatment-Related Adverse Events (TRAE) with incidence $\geq 15\%$ has been reported with CTCAE grade for the safety set (supplementary table 3).

I would suggest adding to the supplementary file a trial checklist for unbiased and transparent reporting of nonrandomized clinical trials (e.g., https://www.cdc.gov/trendstatement/pdf/trendstatement_TREND_Checklist.pdf)

Regarding references, when referring to HER2 centralized assessment, the guidelines utilized should be mentioned (e.g., Wolff et al. Human Epidermal Growth Factor Receptor 2 Testing in Breast Cancer: ASCO-College of American Pathologists Guideline Update. *Journal of Clinical Oncology* 2023 41:22, 3867-3872 or, alternatively, the older version used). The other suggested references have been

Reviewer #2:

Remarks to the Author:

The revised manuscript has addressed all my concerns. FS-1502 has shown efficacy and safety in advanced HER2-Expressing breast cancer. FS-1502 could be a new option for advanced HER2-Expressing breast cancer patients in the future.

Reviewer #3:

Remarks to the Author:

Thank you for your revisions.

Responses to reviewers' comments

Dear reviewer,

On behalf of my co-authors, I would like to thank you for considering our manuscript “HER2-targeting antibody drug conjugate FS-1502 in HER2-expressing metastatic breast cancer: a phase 1a/1b trial” (Manuscript # NCOMMS-23-44187-T) for publication in *Nature Communications*. We have revised the manuscript and have provided a point-by-point response to the reviewers' comments below.

Reviewer #1 (Remarks to the Author):

This is a phase 1a/b (dose escalation and dose expansion) multi-center clinical trial. The trial enrolled patients with HER2-expressing solid tumors (n=5) and specifically HER2-positive metastatic breast cancer (n = 145), in the dose escalation and dose expansion parts, respectively. The primary endpoints of the study were Dose-Limiting Toxicities (DLTs) for the dose-escalation part and Overall Response Rate (ORR) for the dose-expansion part. At the data cutoff, 1 DLT each was reported at 3.0 and 3.5 mg/kg, with no MTD defined, and a RP2D of 2.3 mg/kg once every 3 weeks selected.

The title and abstract sections have been revised according to the reviewers' suggestions. Specifically, the trial registration number has been added, as well as more detailed safety information (e.g., pneumonitis). In addition to best ORR, the authors added confirmed Partial Response (PR). The title now states “HER2-expressing” tumors, which better aligns with a trial that included both a dose escalation and a dose expansion phase, with other tumors allowed in the former phase, in addition to the fact that the trial involved both HER2-positive and HER2-low tumors.

Response:

Thank you for the confirmation.

In the introduction, I appreciate the authors for adding concise introductory information about antibody-drug conjugates as a drug class, with a brief description of their structure. A brief description of the positioning of currently used HER2-directed ADCs in the management of patients with metastatic breast cancer has also been added as required. I would suggest a more precise wording regarding the clinical indication for T-DXd and T-DM1. T-DXd has been approved in unresectable/metastatic HER2-positive breast cancer who received a prior anti-HER2-based regimen either in the metastatic setting or in the (neo)adjuvant setting and have developed disease recurrence during or within six months of completing therapy; it has been also approved in unresectable/metastatic HER2-low (i.e., IHC 1+ or 2+/ISH-) breast cancer who have received a prior chemotherapy in the metastatic setting or developed disease recurrence during or within 6 months of completing adjuvant chemotherapy. I suggest also improving the indication for T-DM1. Indeed, after the positive results in terms of median Progression-Free Survival (mPFS) and Overall Survival (OS) of HER2CLIMB (tucatinib-capecitabine-trastuzumab) and DESTINY-Breast03 (T-DXd), T-DM1 is now recommended in the metastatic setting after progression on either/both regimens by international medical oncology guidelines.

Response:

Thank you for the suggestion. We have added clear descriptions for approved indications of T-DXd and T-DM1 in the Introduction.

Clean file Page 5-6, lines 75-89:

T-DXd is a HER2-directed antibody and topoisomerase inhibitor conjugate and has been approved for the treatment of adult patients with unresectable or metastatic HER2-positive breast cancer who have received a prior anti-HER2-based regimen either in the metastatic setting or in the (neo)adjuvant setting and have developed disease recurrence during or within 6 months of completing therapy; it has also been approved for the treatment of adult patients with unresectable or metastatic HER2-low (IHC 1+ or 2+/ISH-) breast cancer who have received prior chemotherapy in the metastatic setting or developed disease recurrence during or within 6

months of completing adjuvant chemotherapy¹⁶. T-DM1, a HER2-targeted antibody and microtubule inhibitor conjugate, has been approved for the treatment of patients with metastatic HER2-positive breast cancer who previously received trastuzumab and a taxane, separately or in combination, where patients should have either received prior therapy for metastatic disease, or developed disease recurrence during or within 6 months of completing adjuvant therapy; T-DM1 has also been approved for the adjuvant treatment of patients with HER2-positive early breast cancer who have residual invasive disease after neoadjuvant taxane- and trastuzumab-based treatment¹⁷.

I appreciate the authors addressing the major concerns related to the method section. The patient population paragraph has been enriched with information about eligibility criteria for each cohort. The authors improved the description of the formal characteristics of this clinical trial. Specifically, they noted that this is a multicenter, open-label, single-arm, dose-escalation and dose-expansion phase 1 trial (NCT03944499) conducted at nine sites in China (section: Study design and treatment). The authors clarified that solid tumors were allowed for the dose escalation part, while only patients with metastatic HER2+ breast cancer were eligible for the dose-expansion part. The primary endpoints for each phase of the study (dose escalation and dose expansion) have been better explained, as well as the study cutoff date for safety data analysis and median follow-up. In the study design section, the rules for dose escalation have been specified.

Response:

Thank you for the confirmation.

The Study design section has been enriched with information regarding the main rules for assessing safety and determining the RP2D, as required. Specifically, if one of three patients in a dose group experienced a DLT, an additional three patients were added to that dose group (3+3 design). If none of the newly added patients out of the total six had a DLT (1/6), the dose would be escalated to the next level. However, if two or more patients out of the total six had a DLT ($\geq 2/6$), enrollment in that dose group would be stopped, and escalation to the next dose level would not be allowed. The subsequent dose increase ratio was 33% until MTD or RP2D. Dose modification rules have been added to the supplementary files. HER2 assessment has been described in the revised version.

Response:

Thank you for the comments.

The 'End points' section has been restructured. Instead of a previous list of endpoints, the authors now provide a concise definition in brackets for each primary safety endpoint of the dose-escalation phase, citing the supplementary appendix for full information (e.g., pages 7-8, marked version).

Response:

Thank you for the comments.

However, in Phase Ia, I suggest describing the endpoints as reported in the attached study protocol. For example, in line 352, ORR is described as “other efficacy endpoints”. In the protocol, the incidence of Dose-Limiting Toxicities (DLTs) is the primary safety endpoint, and ORR is the primary efficacy endpoint. Other efficacy endpoints were only progression-free survival (PFS), overall survival (OS), 1-year OS, Duration of Response (DOR), and clinical benefit rate (CBR) (see page 2143 of the study protocol). Disease Control Rate (DCR) was not pre-specified in the protocol. Endpoint abbreviations should be correctly explained when they first appear in the text. Some of those like ORR, DLT, MTD, RP2D have been described in the abstract. However, CBR should be clinical benefit rate (not response, see line 358). Also, I would suggest making the nomenclature of the secondary safety endpoints (e.g., “any AEs” incidence, “TEAEs”, “drug-related AEs”, “drug-related TEAEs”) more consistent (see protocol versus supplementary tables versus revised version of the manuscript).

Response:

Thank you for the reviewer's suggestion. We have amended the descriptions for ORR and DCR per protocol, and corrected the full name of CBR. We have made the nomenclature of the secondary safety endpoints consistent, such as incidence of TEAEs, drug-related TEAEs, TEAEs leading to drug interruption/reduction.

Clean file Page 16-17, lines 380-403:

The primary end points of the dose-escalation part were DLTs (defined as predefined toxicities that occur during the DLT observation period, details are provided in the Supplementary Information), MTD (defined as the maximum dose of <33% of DLT events observed in patients with evaluable DLT events), RP2D (determined based on analysis of PK/PD, safety and efficacy results). Secondary end points included safety end points other than DLTs, such as incidence of **TEAEs, SAEs, and TEAEs leading to drug discontinuation/death**. Other secondary end points were **ORR (defined as the proportion of patients with confirmed CR and PR according to RECIST version 1.1) assessed by the investigator**, PFS (defined as the time from the first dose of study treatment to disease progression or death, whichever occurred first), OS (defined as the time from the initiation of study treatment to death due to any cause), 1-year OS rate (defined as the proportion of patients that survived within 1 year of the initiation of study treatment), DOR (defined as the time from first CR or PR to disease progression or death due to any cause, whichever occurred first), and **clinical benefit rate (CBR, defined as the proportion of patients with CR, PR and SD lasting ≥24 weeks according to RECIST version 1.1)**, PK parameters of FS-1502, total antibody and unconjugated MMAF (including maximum concentration, half-lives, area under the serum concentration-time curves, clearance and accumulation ratio, etc), ADA and the NAb of FS-1502. **DCR (defined as the proportion of patients with CR, PR, and SD lasting ≥6 weeks) based on the investigator's assessment was not a prespecified end point, but was also analyzed in the efficacy analysis.**

The primary end point of the dose-expansion part was to evaluate the IRC-assessed ORR of patients with HER2-positive breast cancer. Secondary end points included safety (**such as incidence of TEAEs, SAEs, TEAEs leading to drug discontinuation, and frequency and cause of death within 30 days after the last dose**), PFS, OS, 1-year OS rate, DOR, and CBR, PK parameters of FS-1502, total antibody and unconjugated MMAF, the ADA and the NAb of FS-1502. **DCR, not a prespecified end point, was also analyzed in the efficacy analysis.**

Clean file Page 8, lines 143-150:

As the secondary end points in phase 1a and 1b, **drug-related treatment-emergent adverse events (TEAEs)** of patients at RP2D and total are summarized in Table 2 (adverse events of all dose groups are listed in Supplementary Table 2 and 3).

Overall, **drug-related TEAEs** were observed in 146 (97.3%) of 150 patients. The most common **drug-related TEAEs** were aspartate aminotransferase (AST) increased ($n = 100$, 66.7%), hypokalemia ($n = 77$, 51.3%) and alanine aminotransferase (ALT) increased ($n = 66$, 44.0%). Drug-related TEAEs of grade ≥3 were observed in 51 (34.0%) patients, with the most common events being hypokalemia ($n = 23$, 15.3%) and decreased platelet count ($n = 12$, 8.0%).

Clean file Page 8, lines 162-163:

As noted previously, hypokalemia was a common **drug-related TEAE**, being observed in 77 patients (51.3%), with 23 patients (15.3%) having grade ≥3.

Clean file Page 9, lines 169-176:

Ocular **drug-related TEAEs** were observed in 83 (55.3%) patients, the majority of which were grade 1 or 2, and the most frequent ocular toxicities related to the study drug were dry eye ($n = 36$, 24.0%), keratitis ($n = 27$, 18.0%), and dry eye syndrome ($n = 17$, 11.3%). Grade 3 ocular **drug-related TEAEs** were reported in four (2.7%) patients; two dry eye, one blurred vision and one dry eye syndrome (all receiving 2.3 mg/kg once every 3 weeks). All of the ocular **TEAEs** were reversible, and even the grade 3 toxicity returned to grade ≤1 through the use of

supportive measures such as ocular lubricant, topical antibiotic and an anti-inflammatory agent, and other corneal epithelial recovery interventions.

Clean file Page 12, lines 252-255:

In the phase 1 study of A166, ocular toxicity was the most common **drug-related TEAE**, with grade ≥ 3 events of corneal epitheliopathy (30.9%), blurred vision (18.5%), dry eyes (7.4%) and peripheral sensory neuropathy (6.2%)³⁶.

In Phase 1b, one of the secondary endpoints included in the method section (lines 364-366) is safety. Please, concisely add what safety means here, therefore frequency of AEs, grade and SAEs leading to treatment discontinuation. Specify if a correlation with drug exposure was pre-specified in the protocol [i.e., AEs “drug-related” versus “not-drug-related”] or not [i.e., any AEs since investigational drug started]), then improve nomenclature consistency of AEs as above. Other secondary endpoints were PFS, OS, 1-year OS rate, DOR, CBR, PK parameters of FS-1502, total antibody and unconjugated MMAF, the ADA and the NAB of FS-1502. Please add that another secondary endpoint was frequency and cause of death within 30 days after the last dose. Specify in the method section that DCR was not a pre-specified endpoint in the protocol (see page 2143); then explain that you additionally report that in the results.

Response:

Thank you for the reviewer’s comments. We have added the items of safety endpoints and specified the “drug-related” and “not-drug-related” AEs in study assessments. We have ensured consistent nomenclature for AEs in the manuscript. Frequency and cause of death within 30 days after the last dose has been added, and DCR as a non-prespecified end point has also been clarified for the phase 1b study.

Clean file Page 17, lines 399-403:

Secondary end points included safety (**such as incidence of TEAEs, SAEs, TEAEs leading to drug discontinuation, and frequency and cause of death within 30 days after the last dose**), PFS, OS, 1-year OS rate, DOR, and CBR, PK parameters of FS-1502, total antibody and unconjugated MMAF, the ADA and the NAb of FS-1502. **DCR, not a prespecified end point, was also analyzed in the efficacy analysis.**

Clean file Page 18, lines 427-431:

The causality of a TEAE to the study drug was prespecified; it was considered not drug related if no study drug was used or the time of the AE was not related to the use of study drug or the cause of the AE was otherwise clear; it was considered drug related if there was evidence of the use of study drug and the occurrence of the TEAE was reasonably related in time to the use of study drug.

RECIST version used has been reported now ('end points' and 'study assessments' paragraphs).

Response:

Thank you for the comment.

In the statistical analysis section (line 394), we find: "An interim analysis using the frequentist method was conducted after a certain amount of data was collected in the trial." This sentence looks a bit ambiguous as it is not clear when a "certain amount of data" was collected. The protocol reports that "an interim analysis using Bayesian posterior probability will be conducted when approximately 20 patients have completed 2 tumor assessments. If the predicted Pr (ORR <20%) is >80%, i.e. fewer than 6 responses are observed in 20 evaluable patients, then there is an 80% probability that the drug ORR is lower than standard of care and early discontinuation of the cohort may be considered, otherwise continuing enrollment to approximately 50 patients.

Bounds will be adjusted based on the actual number of people in the Efficacy Analysis Set. The interim analysis margins are non-binding and the sponsor will consider the safety and effectiveness data in the final decision." Moreover, if the highlighted sentence above was added to better specify the main rules for study discontinuation as mentioned in the rebuttal letter, I believe the goal of early discontinuation and pre-specified rules for the analysis should be better explained in the text, as well.

Response:

Thank you for the comments. We have removed the ambiguous description of the interim analysis and provided a summary of protocol V6.0 amendments as the supplement. It is now stated that the frequentist method was used for interim analysis instead of Bayesian posterior probability.

Clean file Page 19, lines 454-455:

An interim analysis using the frequentist method was conducted after data from 65 patients was collected in the phase 1b trial.

In the results section, I would recommend using the wording "patients with breast cancer", instead of breast cancer patients, as it is more inclusive of the instances of patient advocates.

Response:

Thank you for the comments. We have amended this throughout the manuscript as below.

Clean file Page 7, lines 121-122:

For patients **with breast cancer**, 112 (77.2%) were HER2-positive and 32 (22.1%) had tumors with HER2-low expression.

Clean file Page 9, lines 181-183:

Of 67 patients **with HER2-positive breast cancer** treated at the RP2D of 2.3 mg/kg, two (3.0%) patients had complete response (CR; one to be confirmed) and 34 (50.7%) patients had partial response (PR; seven to be confirmed).

Clean file Page 9-10, lines 191-193:

Among 15 efficacy evaluable patients **with HER2-low expression breast cancer** treated at the RP2D of 2.3 mg/kg, four (26.7%) patients experienced PR and five (33.3%) patients experienced SD.

Clean file Page 12, lines 265-267:

As of the data cutoff date, objective responses were observed in patients **with HER2-positive breast cancer** at dose levels of 1.0 mg/kg and higher, and responses were also observed in some patients **with HER2-low breast cancer** and non-breast cancer patients.

Clean file Page 12, lines 272-275:

FS-1502 demonstrated encouraging antitumor activity with the best ORR of 53.7% and the median PFS of 15.5 months (95% CI, 4.6-not reached) for patients **with HER2-positive breast cancer** at the RP2D of 2.3 mg/kg once every 3 weeks, with the median DOR and OS not yet reached due to short follow-up.

Clean file Page 13, lines 287-289:

FS-1502 showed higher ORR in **patients with HER2-positive than those with HER2-low breast cancer**, it may be related to the hydrophobicity and low bystander cell penetration of payload effect of MMAF⁴⁰.

In Supplementary files a new table summarizing Treatment-Related Adverse Events (TRAE) with incidence $\geq 15\%$ has been reported with CTCAE grade for the safety set (supplementary table 3).

Response:

Thank you for the reviewer's confirmation.

I would suggest adding to the supplementary file a trial checklist for unbiased and transparent reporting of nonrandomized clinical trials (e.g., https://www.cdc.gov/trendstatement/pdf/trendstatement_TREND_Checklist.pdf)

Response:

Thank you for the reviewer's suggestion. We have added the TREND checklist as one supplementary file.

Regarding references, when referring to HER2 centralized assessment, the guidelines utilized should be mentioned (e.g., Wolff et al. Human Epidermal Growth Factor Receptor 2 Testing in Breast Cancer: ASCO–College of American Pathologists Guideline Update. Journal of Clinical Oncology 2023 41:22, 3867-3872 or, alternatively, the older version used). The other suggested references have been mostly implemented as required.

Response:

Thank you for the reviewer's comments. We have cited the reference for HER2 assessment.

Clean file Page 15, lines 348-350:

HER2 expression was assessed using immunohistochemistry staining and/or fluorescence in situ hybridization gene amplification testing in tumor tissue sections or fresh tissues⁴².

Reviewer #2 (Remarks to the Author):

The revised manuscript has addressed all my concerns. FS-1502 has shown efficacy and safety in advanced HER2-Expressing breast cancer.FS-1502 could be a new option for advanced HER2-Expressing breast cancer patients in the future.

Response:

Thank you for the confirmation.

Reviewer #3 (Remarks to the Author):

Thank you for your revisions.

Response:

Thank you for the confirmation.